# Long-Term Vemurafenib Exposure Induced Alterations of Cell Phenotypes in Melanoma: Increased Cell Migration and Its Association with EGFR Expression

**DOI:** 10.3390/ijms20184484

**Published:** 2019-09-11

**Authors:** Eszter Molnár, Tamás Garay, Marco Donia, Marcell Baranyi, Dominika Rittler, Walter Berger, József Tímár, Michael Grusch, Balázs Hegedűs

**Affiliations:** 12nd Department of Pathology, Semmelweis University, H-1091 Budapest, Hungary; 2Pázmány Péter Catholic University, Faculty of Information Technology and Bionics, H-1083 Budapest, Hungary; 3Oncology Center, Semmelweis University, H-1091 Budapest, Hungary; 4National Center for Cancer Immune Therapy, Department of Oncology, Herlev and Gentofte Hospital, University of Copenhagen, DK-2730 Herlev, Denmark; 5Institute of Cancer Research, Medical University of Vienna, A-1090 Vienna, Austria; 6Department of Thoracic Surgery, University Medicine Essen-Ruhrlandklinik, University Duisburg-Essen, D-45239 Essen, Germany

**Keywords:** melanoma, V600E BRAF mutation, vemurafenib resistance, EGFR, PD-L1, migration

## Abstract

Acquired resistance during BRAF inhibitor therapy remains a major challenge for melanoma treatment. Accordingly, we evaluated the phenotypical and molecular changes of isogeneic human V600E BRAF-mutant melanoma cell line pairs pre- and post-treatment with vemurafenib. Three treatment naïve lines were subjected to in vitro long-term vemurafenib treatment while three pairs were pre- and post-treatment patient-derived lines. Molecular and phenotypical changes were assessed by Sulforhodamine-B (SRB) assay, quantitative RT-PCR (q-RT-PCR), immunoblot, and time-lapse microscopy. We found that five out of six post-treatment cells had higher migration activity than pretreatment cells. However, no unequivocal correlation between increased migration and classic epithelial–mesenchymal transition (EMT) markers could be identified. In fast migrating cells, the microphthalmia-associated transcription factor (MITF) and epidermal growth factor receptor (EGFR) mRNA levels were considerably lower and significantly higher, respectively. Interestingly, high EGFR expression was associated with elevated migration but not with proliferation. Cells with high EGFR expression showed significantly decreased sensitivity to vemurafenib treatment, and had higher Erk activation and FRA-1 expression. Importantly, melanoma cells with higher EGFR expression were more resistant to the EGFR inhibitor erlotinib treatment than cells with lower expression, with respect to both proliferation and migration inhibition. Finally, EGFR-high melanoma cells were characterized by higher PD-L1 expression, which might in turn indicate that immunotherapy may be an effective approach in these cases.

## 1. Introduction

Melanoma is the most deadly skin cancer [1]. However, in the past few years, several treatments have been approved for the treatment of BRAF-mutant metastatic melanoma. Vemurafenib was the first selective BRAF inhibitor that showed dramatic therapeutic responses in patients with V600 BRAF-mutant melanoma [2]. Later, dabrafenib and encorafenib were also clinically approved to target V600 BRAF-mutated tumors [3,4]. Furthermore, combination treatment of V600 BRAF-specific inhibitors with MEK inhibitors showed higher progression-free and overall survival rates compared with BRAF inhibitors alone [5]. Despite the initial success of these targeted therapies, clinical benefit is usually transient due to the rapid emergence of drug resistance [6]. Resistance mechanisms against BRAF inhibitors include point mutations in other proteins (e.g., RAS, MEK, AKT, PI3K, CDK4/6), tumor suppressor deletions (PTEN), amplifications or alternative splicing in BRAF, or increased expression and activation of receptor tyrosine kinases such as EGFR, IGF1R, or PDGFR [7,8,9,10,11,12,13,14,15]. Also, resistant cells have elevated PD-L1 expression, which could induce a more aggressive phenotype of melanoma cells [16]. Several studies indicate that BRAF-inhibitor-resistant cells have a more invasive phenotype than BRAF-inhibitor-sensitive cells, which is in part attributed to epithelial to mesenchymal transition (EMT) [8,17,18,19]. EMT of melanoma cells is under regulation of transcription factors (Zeb, Snail, Twist) that in turn are affected by several signaling pathways, like the RAS/RAF/MEK/ERK, PI3K/AKT/mTOR, Wnt/ß-catenin, or TGF-ß pathways [20]. Importantly, in advanced melanoma, oncogenic BRAF and NRAS mutation can modulate EMT through the MEK/ERK/FRA-1 pathway [21,22]. In certain tumor types, an important marker of EMT is loss of E-cadherin expression and increase of *N*-cadherin expression. In V600 BRAF-mutant melanoma cells, loss of E-cadherin occurs frequently and it could be responsible for higher metastatic activity [23]. Furthermore, the expression of vimentin and matrix metalloproteases often increases to provide a more invasive phenotype [21,24]. The MITF transcription factor is also an important factor regulating EMT in melanoma. Low MITF expression leads to an invasive phenotype, while high MITF expression results in differentiation and a more proliferative phenotype [22,25].

The aim of our study was to explore melanoma proliferation and cell migration and molecular changes upon long-term vemurafenib treatment. To this end, isolated cell cultures from biopsies taken from patients before and during their treatment (Mel KD, Mel JL, Mel JR) and also treatment naïve cell lines which were treated long-term with vemurafenib in vitro (MM90906, MM90911, MM040111) were investigated [26,27]. We found that in preclinical models of the long-term vemurafenib-exposed melanoma cells, high EGFR expression could relate to high migratory but not proliferative capacity, vemurafenib and erlotinib resistance, and also elevated PD-L1 expression.

## 2. Results

### 2.1. Proliferation and Migration of the Melanoma Cell Line Pairs

In order to characterize the phenotype of melanoma cell line pairs, vemurafenib IC50 value, proliferation, and migration of the cells were evaluated via SRB assay and time-lapse microscopy, respectively. Three cell lines (Mel KD, Mel JL, MM90906) were intrinsically resistant to vemurafenib, while three cell lines (Mel JR, MM90911, MM040111) were initially sensitive to vemurafenib but their post-treatment vemurafenib IC50 values increased remarkably (Table 1). Proliferation capacity of the cell lines increased in only two out of the six cases (Mel JR, MM90906) (Figure 1A). In contrast, migration of the cell lines increased significantly in five out of six cell lines. In the MM90906 cell line pair, pretreatment migration activity was extremely high and did not significantly increase further after vemurafenib treatment (Figure 1B).

### 2.2. Evaluation of EMT Marker Expression by q-RT-PCR

Next, we further investigated the potential molecular background of the increased migration activity. mRNA levels of “classic” EMT markers (SNAIL, ZEB-1, MMP-3, MMP-1, vimentin, *N*-cadherin, *E*-cadherin) of the transcription factors MITF and FRA-1 and of the epidermal growth factor receptor (EGFR) were determined via q-RTPCR. Since migration activity was elevated in all cell line pairs, we expected to see similar changes in EMT markers in all post-treatment cell lines. However, cell line pairs showed considerable variability in expression changes of EMT markers. Of note, both in vivo (Mel) and in vitro (MM) established cell lines were characterized with heterogeneous alterations. Also, we found no E-cadherin expression in any of the cell lines either pre- or post-treatment (Figure 2A; Appendix A). Only vimentin showed an increase in all cell line pairs except for Mel JR. EGFR mRNA expression level of the cells was also determined and, again, showed an increase in all cell lines except for Mel JR. Since we found no considerable correlation with respect to the pre- and post-treatment dichotomization, we correlated the mRNA expression levels of the investigated factors with proliferation and migration of the cells. Cut-off values for proliferation and migration indices were 10 and 50, respectively (for further details see Section 4). We found that MITF mRNA expression was considerably (*p* = 0.075) lower, while EGFR mRNA expression was significantly (*p* = 0.016) higher in fast migrating melanoma cells (Figure 2B). Furthermore, in cells with high proliferative capacity, FRA-1 mRNA level was significantly (*p* = 0.037) lower than in slowly proliferating cells (Figure 2C; Appendix A).

### 2.3. Signaling Pathway Activation and EGFR, PTEN, MITF, FRA-1, and PD-L1 Expression of the Cell Line Pairs

EGFR, MITF, FRA-1 expression was further analyzed on protein level. MAPK and PI3K/AKT signaling pathway activations were characterized by pErk/Erk and by pAkt (Ser473)/Akt ratio, respectively (Figure 3). In the majority of cell line pairs, there was no significant difference in Erk activation upon long-term vemurafenib treatment. Interestingly, Erk activation significantly decreased in post-treatment Mel JR cells, while it increased in post-treatment Mel JL cells (Appendix A). However, Akt activation changed in almost all cell line pairs. In Mel KD and Mel JR cells, Akt activation was significantly decreased; in Mel JL, MM90906, and MM90911, it was significantly increased (Appendix A). We found a decrease in PTEN expression in Mel JL and two pairs had overall reduced (Mel KD) or completely lost (MM909011) PTEN expression. Furthermore, the pAkt/Akt ratio tended to be higher in PTEN-low cells (Appendix A). Importantly, EGFR expression notably increased in all post-treatment cell lines except for Mel JR, in line with findings at the transcriptional level (Figure 4A). Next, we dichotomized the cell line panel to EGFR-low (Mel JL pre, Mel JR pre, Mel JR post, MM909011 pre, MM040111 pre, MM040111 post) and EGFR-high (Mel KD pre, Mel KD post, Mel JL post, MM90906 pre, MM90906 post, MM909011 post) groups (Figure 4B and Appendix A). EGFR-high cells tended to be more resistant to vemurafenib (*p* = 0.029) and also had higher migration (*p* = 0.042) but not proliferation (*p* = 0.266) index than EGFR-low cells. Furthermore, in EGFR-high cells, there was a considerably higher pErk/Erk ratio (*p* = 0.003) and FRA-1 (*p* = 0.055) expression. However, MITF expression did not correlate with EGFR expression on the protein level. Also, low MITF expression in highly migrating cells could not be further confirmed on the protein level.

Finally, we evaluated PD-L1 protein level in cell line pairs since anti-PD-1 immunotherapy is an important therapeutic approach in melanoma. We found that in EGFR-high melanoma, there is a significantly (*p* = 0.029) higher PD-L1 expression than in EGFR-low melanoma cells (Figure 4B).

### 2.4. High-EGFR-Expressing Cells Are More Resistant to Erlotinib Treatment

Since EGFR expression showed positive correlation with Erk activation, we tested the EGFR inhibitor erlotinib in our panel of melanoma cells. Interestingly, cell lines with high EGFR expression were significantly more resistant to erlotinib than cells with low EGFR expression in five-day viability assays (Figure 5A,B). Additionally, the four fastest cell lines (MM90906 pre, MM90906 post, MM90911 post, and MM040111 post) were treated with erlotinib and changes in migration activity were determined. We found that erlotinib treatment lowered migration activity in all cell lines, however, it showed the most robust migration inhibition on the EGFR-low MM040111post cell line (Figure 5C).

## 3. Discussion

Our aim was to investigate the effects of long-term vemurafenib treatment on V600E BRAF-mutant pre- and post-treatment melanoma cell line pairs. In four out of six cell line pairs, proliferation capacity decreased (Mel KD, Mel JL, MM90911, MM040111), while in two cell pairs, proliferation capacity increased (Mel JR, MM90906) in post-treatment cells (Figure 1A). Migration, in contrast, increased in all cell line pairs upon vemurafenib treatment (Figure 1B). Regarding the molecular background of the increased migration, we hypothesize that EMT-associated factors might be altered. E-cadherin loss occurs frequently in V600E BRAF-mutant melanomas [23]. This phenomenon was further confirmed in our work, since all cells had lost E-cadherin expression even before vemurafenib exposure. Based on previous data, we expected to see elevated mRNA expression levels of EMT markers. However, we could not detect a clear pattern in our cell line pairs (Figure 2A). Only vimentin expression increased in all post-treatment cells except for Mel JR. Furthermore, Cordaro et al. found that dabrafenib-resistant cells increase their motility, despite the EMT markers’ expression remaining unaltered (E-cadherin, Snail) or even decreased (Twist), which suggests a “distinct active EMT-like” process adopted by melanoma cells under drug exposure [17]. Besides classic EMT markers, we also evaluated mRNA levels of FRA-1 and MITF transcription factors—both of which are important factors of melanoma progression—and their correlation with migratory and proliferative capacity. MITF mRNA expression showed negative correlation with migration and also in four out of six cases, MITF mRNA level decreased in post-treatment cells. It is in line with previous observations that low MITF expression leads to a more invasive phenotype [25,28]. Also, silencing MITF expression in melanoma cells has been previously shown to result in faster migration [29]. Furthermore, in a retrospective study, it was demonstrated that low MITF expression is associated with a worse prognosis in melanoma patients [30]. We found that in fast migrating cells, EGFR mRNA expression is significantly higher than in slow migrating cells. Furthermore, FRA-1 mRNA expression revealed a negative correlation with proliferation. Moreover, in post-treatment cells with reduced proliferation activity (Mel KD, Mel JL, MM90911, and MM040111), FRA-1 mRNA expression increased (three out of four cases), whereas in post-treatment cells with elevated proliferation activity (Mel JR and MM90906), FRA-1 mRNA expression decreased. FRA-1 overexpression was previously shown to inhibit proliferation in glioma cells [31]. However, in breast cancer cells, FRA-1 overexpression could elevate proliferation activity [32]. Obenauf et al. investigated the impact of FRA-1 expression in a melanoma model and they found that adding BRAF inhibitor to sensitive cells resulted in FRA-1 expression decrease. The “therapy-induced secretome” generated upon FRA-1 repression could increase proliferation and metastatic capacity of BRAF-inhibitor-resistant cells via the PI3K/Akt pathway [33,34]. However, our data suggest that high FRA-1 mRNA expression could relate to lower proliferative capacity of melanoma cells. At the protein level, only the positive correlation between EGFR expression and migration activity was confirmed. We also found elevated EGFR expression upon long-term vemurafenib exposure in five out of six pairs. Importantly, cells with higher IC50 value for vemurafenib had higher EGFR expression. EGFR-activation-mediated BRAF inhibitor resistance was described first in V600E BRAF-mutant colorectal cancer [35]. Activation of the EGFR pathway was also identified in BRAF-inhibitor-resistant melanoma [7,8,36]. The EGFR pathway could drive proliferation and metastatic activity of the cells [7]. However, a recent study showed that upregulation of certain genes (e.g., EGFR, ANGPLT4, VCAM-1) upon BRAF inhibitor treatment is associated with elevated cell migration and angiogenesis [37]. We also found that higher EGFR expression indicates elevated migration but not proliferation of melanoma cells. In light of these findings, we also tested the EGFR inhibitor erlotinib in our panel of melanoma cells. In lung cancer, high EGFR expression in EGFR wild-type patients was shown to be a positive predictive factor for erlotinib treatment [38]. Interestingly, we found that cells with high EGFR expression were more resistant to erlotinib treatment than cells with low EGFR expression. Also, migration inhibition with erlotinib was more profound in EGFR-low cell lines. Interestingly, it was also shown in colorectal cancer that lower EGFR expression could be associated with better efficacy of anti-EGFR therapy [39,40]. Recent clinical trials showed that combined EGFR + BRAF + MEK inhibition is tolerable, with promising activity in patients with BRAF V600E-mutant colorectal patients [41,42]. Also, a previous preclinical study suggests that EGFR inhibitors may have different effects on melanoma cells depending on their mutational status, and that EGFR TKI-s especially in combination could be an effective approach to eliminate BRAF-mutant melanoma [43]. Our findings also could implicate that EGFR + BRAF + MEK combination therapy might improve efficacy of targeted therapy also in melanoma patients. Of note, our data also revealed that EGFR-high melanoma cells express more PD-L1 than EGFR-low cells. It has already been shown in lung cancer that EGFR pathway activation could induce PD-L1 expression [44]. Of note, in melanoma, PD-L1/PD-L2/JAK2 amplification could be associated with durable response to immunotherapy [45]. Furthermore, recently, a novel EGFR/PD-L1 bispecific antibody was developed to enhance efficacy of PD-L1 inhibition in EGFR-overexpressing tumors [46].

Altogether, our study demonstrated that long-term vemurafenib exposure in BRAF-mutant melanoma can lead to increased migration that correlates with elevated EGFR expression. Furthermore, high levels of EGFR are associated with a lower sensitivity against BRAF and EGFR inhibitors but the increased PD-L1 expression may potentiate immune-oncological therapies.

## 4. Materials and Methods

### 4.1. Cell Lines and Reagents

Mel KD, Mel JL, and Mel JR cell line pairs were gifted from Peter Hersey, Oncology and Immunology Unit, Calvary Mater Newcastle Hospital and Kolling Institute, Royal North Shore Hospital, University of Sydney, New South Wales, Australia (Roche/Plexxikon NP22657, the trial was approved by the Hunter Area Research Ethics Committee) [27]. MM90906, MM90911, and MM040111 cell lines pairs were established at the National Center for Cancer Immune Therapy, Herlev Hospital, Denmark. Mutation status and patient data for Mel KD (Patient 1), Mel JL (Patient 4), and Mel JR (Patient 3) cells are available in Lai et al., 2012 [27]. BRAF-mutation status for MM90906, MM90911, and MM040111 was tested with pyrosequencing, using primers as described by Richman et al. [47]. MM90906 (Patient 5), MM90911 (Patient 4), and MM040111 (Patient 3) cell lines were described previously in Donia et al., 2012 [26]. MM90906, MM909011, and MM040111 were treated in vitro with vemurafenib in increasing concentration (0.001–10 µM). All cell lines were established from metastatic sites [26,27]. All cell lines contained TERT promoter mutation (own sequencing). Mel KD, Mel JR, and MM90911 had C124T; Mel JL, MM90906, and MM040111 had C146T TERT promoter mutation. Cells were cultured in DMEM (Lonza, Switzerland) supplemented with 10% fetal bovine serum (EuroClone, Pero, Italy) and 1% penicillin–streptomycin–amphotericin (Lonza, Switzerland) at 37 °C with humidified 5% CO_2_ atmosphere. Vemurafenib and erlotinib were purchased from Selleck Chemicals (Houston, TX, USA).

### 4.2. Cell Proliferation Assay

Cells were seeded in the inner 60 wells of 96-well plates. After treatment, 10% trichloroacetic acid was added to each well and stained for 15 min with SRB. Excess stain was discarded and cells were washed with 1% acetic acid solution. Stained cells were dissolved in 10 mM Tris-HCl pH 8 and OD was measured at 570 nm using a microplate reader (EL800, BioTec Instruments, Winooski, VT, USA). Relative cell growth was calculated by comparison of the A570 reading from treated versus control wells. To determine proliferation rate, cells were fixed and stained as described previously on day 1, 3, 5, and 7 after plating. Proliferation index was calculated by day5/day1 A570 reading. Cut-off value was 10 for low and high proliferating cells.

### 4.3. Time-Lapse Microscopy

Time-lapse microscopy measurements were implemented and evaluated as described previously [48]. Briefly, cells were tracked for 24 h using 10 min interval images and for each time interval, average displacement was calculated. We evaluated displacement of at least seven individual cells per microscopic field with a manual marking cell-tracking program and recorded position parameters of the tracked cells into data files. Migration index was determined by averaging for each tracked cell the net displacement for all 20 h intervals in at least three independent experiments and three microscopic fields. Cut-off value was 50 µm displacement in 20 h for slow and fast migrating cells.

### 4.4. Gene Expression Analysis

Total RNA was isolated from cells with TRISOL reagent (guanidine thiocyanate 0.9 M, ammonium thiocyanate 0.45 M, sodium acetate 0.12 M, glycerine 6%, phenol saturated with water 39%, and RNAse-free water). cDNA was prepared using the Revert Aid reverse transcriptase (Thermo Fisher Scientific, Waltham, MA, USA). A 2 ug amount of RNA was used to reverse transcriptase reaction. Changes in gene-specific transcript copy number were evaluated using qPCR with Maxima SYBR/ROX master mix (Thermo Fisher Scientific, Waltham, MA, USA) and Maxima Probe/ROX master mix (Thermo Fisher Scientific, Waltham, MA, USA). The gene expression levels were measured against the housekeeping gene GAPDH. Primer sequences are available in the Appendix A. HeatMap was designed with MatLab software (Natick, MA, USA).

### 4.5. Immunoblot

For the immunoblot analysis, 6% trichloroacetic-acid-precipitated cell pellets were dissolved in modified Läemmli-type sample buffer containing 90 mM Tris-HCl, pH 7.9, 2% sodium dodecyl sulfate (SDS), 10% glycerol, 5 mM ethylenediaminetetraacetic acid (EDTA), 125 mg/mL urea, 100 mM dithiothreitol (DTT), 0.02% bromophenol blue. The protein concentration of the samples was determined with Qubit 4 Fluorimeter according to the manufacturer’s instructions (Thermo Scientific, Waltham, MA, USA). Cell lysates were subjected to sodium dodecyl sulfate-polyacrylamide gel electrophoresis (SDS-PAGE). Protein amount was 20 ug/lane in the immunoblot assays. Primary antibodies were used for p-Erk1/2/Erk1/2, p-Akt (Ser473)/Akt, PTEN, EGFR, FRA-1, PD-L1, and MITF (Cell Signaling #9101, #9102, #4058, #9272, #9188, #4267, #5281, #13684, and Dako, M3621, respectively) and as loading control anti-GAPDH (Cell Signaling #5174). Signals were developed using ECL Star Enhanced Chemiluminescent Substrate (EuroClone, Pero, Italy). The bands were quantified by densitometry using ImageJ software [49] and normalized using the expression levels of GAPDH.

### 4.6. Statistics

All statistical analyses were performed in GraphPad Prism 5 (GraphPad Software Inc, San Diego, CA, USA). To evaluate significant differences between groups, one-way ANOVA followed by Tukey’s post hoc test was performed. In case of two groups unpaired t-tests were applied. Differences were considered significant at *p* < 0.05.

## Figures and Tables

**Figure 1 ijms-20-04484-f001:**
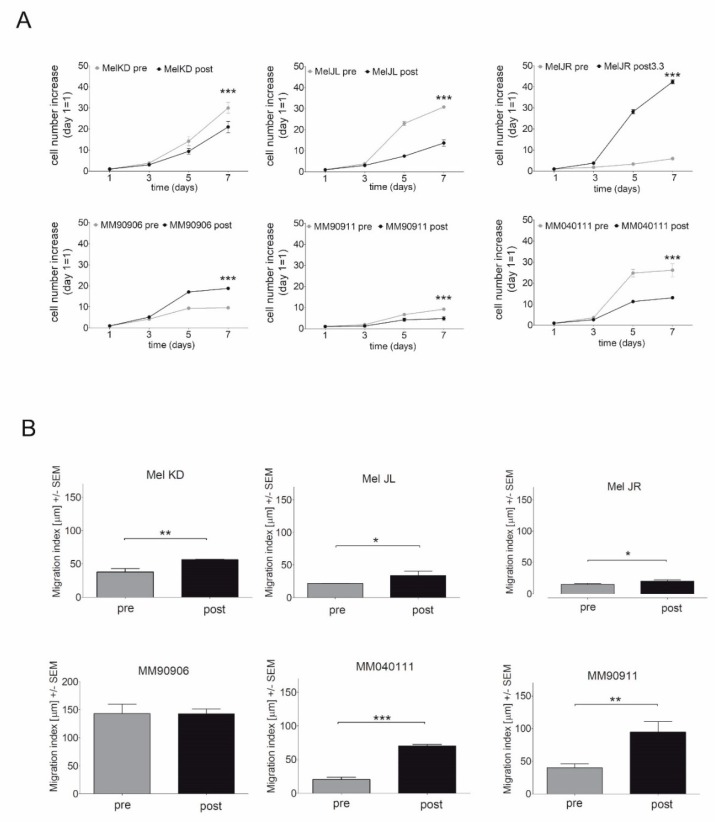
Proliferation and migration of melanoma cells. (**A**) Relative proliferation rate of cell line pairs (mean values +/− SEM). (**B**) Migration index is equivalent to the average net displacement of the cells in 20 h (mean values +/− SEM) (* *p* ≤ 0.05, ** *p* ≤ 0.01, *** *p* ≤ 0.001).

**Figure 2 ijms-20-04484-f002:**
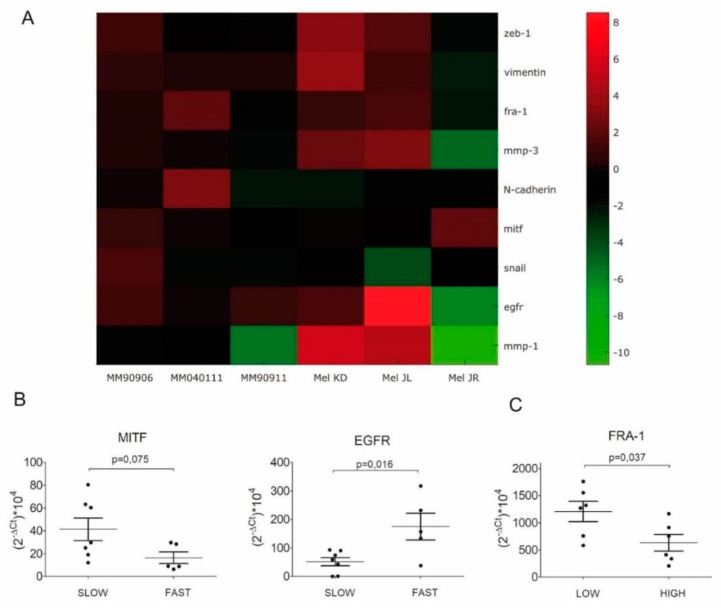
mRNA expression of EMT markers, MITF, FRA-1, and EGFR of cell line pairs. (**A**) Heatmap of mRNA expression. Green indicates repressed mRNA levels and red elevated levels. GAPDH was used as housekeeping gene. (**B**) In fast migrating cells, there was considerably lower (*p* = 0.075) MITF and significantly higher (*p* = 0.016) EGFR mRNA expression. Cut-off value was 50 µm displacement in 20 h for dichotomizing slow and fast migrating cell lines. (**C**) Significantly lower (*p* = 0.037) FRA-1 mRNA expression was measured in highly proliferating cells.

**Figure 3 ijms-20-04484-f003:**
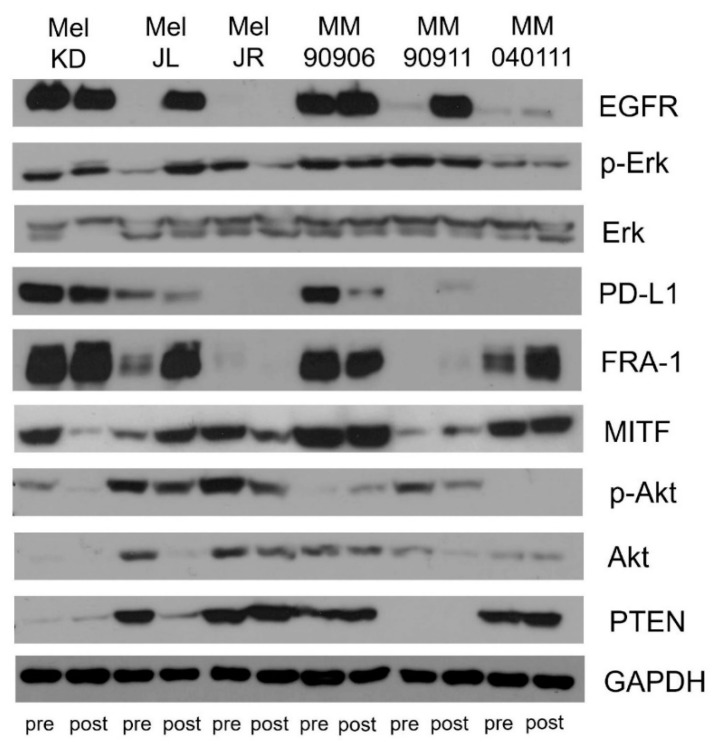
Immunoblot analysis of pErk/Erk, pAkt (Ser473)/Akt, EGFR, MITF, FRA-1, PTEN, PD-L1 expression of the cell line pairs. Blots are representative images from three independent experiments.

**Figure 4 ijms-20-04484-f004:**
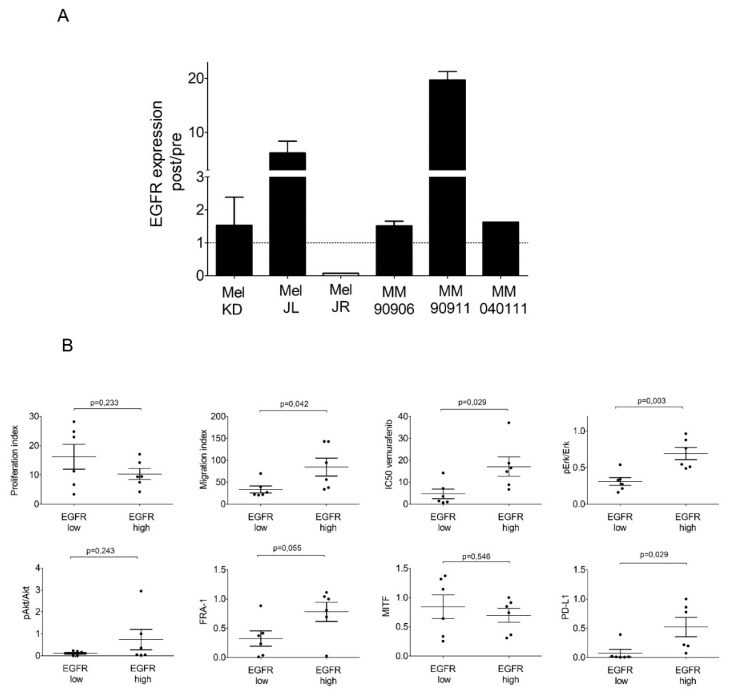
The impact of EGFR expression in V600E BRAF-mutant melanoma cells. (**A**) EGFR expression was elevated in five out of six cell line pairs. (**B**) EGFR-high melanoma cell lines showed significantly higher migration index, vemurafenib IC50 values, pErk/Erk ratio, and FRA-1 and PD-L1 expression.

**Figure 5 ijms-20-04484-f005:**
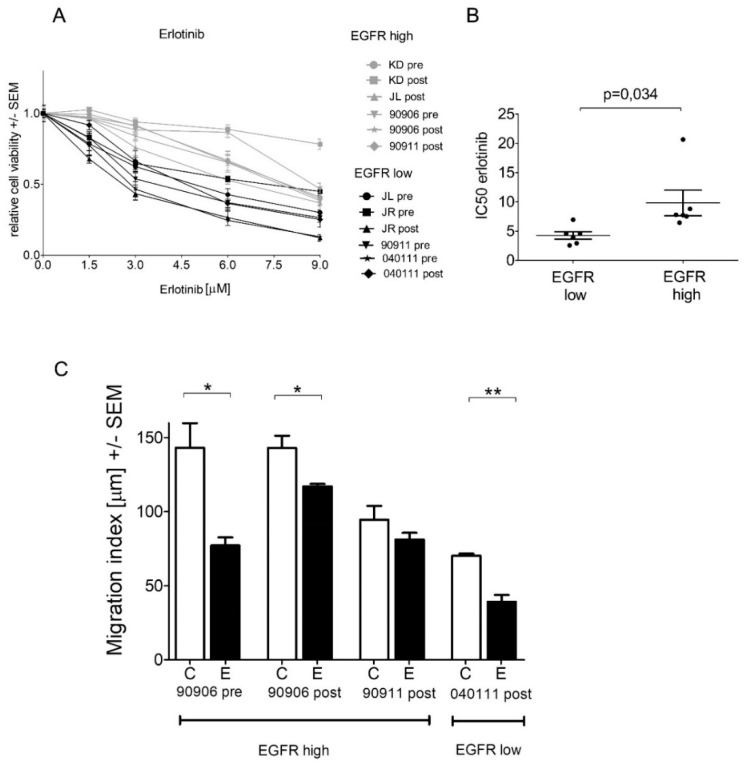
Erlotinib treatment of cell line pairs. (**A**) Growth inhibition effect of erlotinib in five days. (**B**) IC50 erlotinib values in EGFR-low and EGFR-high cells. (**C**) Migration inhibition effect of erlotinib on the four fastest cell lines. Migration index was determined between 48 and 68 h after the erlotinib treatment. C = control, E = erlotinib (6 μM). Data is shown as the mean +/− SEM (* *p* ≤ 0.05, ** *p* ≤ 0.01).

**Table 1 ijms-20-04484-t001:** IC50 values and patient data of the cell line pairs. (M = male, F = female, PR = partial response, n.a. = not applicable).

Cell Line	Pre-TX IC50 [µM]	Post-TX IC50 [µM]	Sex	Age	Response to Vemurafenib	Reference
Mel KD	6.7	18.6	M	35	PR	Patient 1 in [27]
Mel JL	14.2	16.5	F	53	PR	Patient 4 in [27]
Mel JR	1	7.1	F	54	PR	Patient 3 in [27]
MM90906	8.8	14.8	M	36	n.a.	Patient 5 in [26]
MM90911	1.5	37.1	M	41	n.a.	Patient 4 in [26]
MM040111	0.7	3.5	F	78	n.a.	Patient 3 in [26]

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
