# Peer review of "Long-Term Vemurafenib Exposure Induced Alterations of Cell Phenotypes in Melanoma: Increased Cell Migration and Its Association with EGFR Expression"

_ijms, 2019, doi:10.3390/ijms20184484_

Round 1

Reviewer 1 Report

I have read a manuscript by TE. Molnar and coauthors titled: Long-term vemurafenib exposure induced alterations of cell phenotypes in melanoma: cell migration and the role of EGFR expression submitted to “International Journal of Molecular Sciences”

The manuscript has some merits which make it potentially interested for readers of IJMS, nevertheless must be improved before publishing.

Title:

Should be more specific

Abstract:

Must be improved, it is too general. More specific data (results) must be presented including migration activity and gene expression data. Objectives should be clearer.

Methods:

It is not clear w migration was calculated? This issue must be explained, how cell trajectories were measured and calculated. GAPDH as a housekeeping gene is absolutely inadequate to melanoma cell line in the context of anticancer treatment. This gene is upregulated in most cancer cell lines [Int J Oncol. 2017 Jan;50(1):252-262; Chemistry & Biology 2014; 21 (11) 1533-1545). Look also Figure 3 (WB for GAPDH). The other normalization method for aRT-PCR should be considered. Which protein concentrations were used for WB? Also the same for a reverse trancriptase (RT) reaction?

Results:

What is the main finding of this study? Erlotinib sensitivity or vemurafenib resistance in the high-EGFR expression cell lines? The regression model for correlation between EGFR and up- and down-regulated gens should be proposed. Figure 5A is not clear.

Discussion:

What is the main finding of this study? See the objectives.

Minor remarks:

Vocabulary

“Migratory capacity” is not adequate to cell behavior, I suggest “migration activity”

“Videomicroscopy” sounds anachronism “time-lapse microscopy “sounds better.

Reviewer 2 Report

This is a thoughtful well written paper examining the molecular changes associated with

 BRAF inhibitor therapy in 6 cell lines. Although the paper would be strengthened by more than 6 cell lines, the number is sufficient to draw appropriate conclusions. The paper would also be strengthened by the following aspects:

Please provide some additional discussion regarding the implication of the results and BRAF treatment protocols.

Please explain more regarding the patient's background and clinical status of the patient biopsies.

It was stated that the Mel KD, Mel JL, and MM90906 were intrinsically resistant. Please explain why were these 3 cell lines intrinsically resistant, how was this determined, what criteria were used, and how does this impact the data/results.

In figure 2 the data were divided into SLOW and FAST moving. Please further define SLOW and FAST moving cell line pairs (i.e., what criteria was used to define) in the data and in the figure legend.

Please define the BRAF therapy received by the patients. Please discuss if a concentration dependence was examined and the impact of higher or lower BRAF therapies.

Please provide further discussion/explanation on what it biologically means that only vimentin expression was increased in all cell lines tested.

Author Response

Reviewer 2 comments: 

- Please provide some additional discussion regarding the implication of the results and BRAF treatment protocols.

Discussion section have been implemented with information about EGFR+BRAF+MEK combination treatment (Discussion section, line 220-226, page 9).

- Please explain more regarding the patient's background and clinical status of the patient biopsies.

We included gender, age and response to vemurafenib treatment data in Table 1. and the Methods section (line 247-248, page 10).

- It was stated that the Mel KD, Mel JL, and MM90906 were intrinsically resistant. Please explain why were these 3 cell lines intrinsically resistant, how was this determined, what criteria were used, and how does this impact the data/results.

Based on previous report on in vitro vemurafenib treated V600E BRAF mutant melanoma cells (Wong et al. 2014) we considered cells vemurafenib resistant above IC50 >2 uM. Mel KD pre, Mel JL pre and MM90906 pre cells had vemurafenib IC50 values 6.7; 14.2 and 8.8 uM, respectively which implicates that these cells were intrinsically resistant to vemurafenib. Of note, Mel KD pre and MM90906 pre cells have elevated EGFR expression which is considered to be a potential factor for vemurafenib resistance.

- In figure 2 the data were divided into SLOW and FAST moving. Please further define SLOW and FAST moving cell line pairs (i.e., what criteria was used to define) in the data and in the figure legend.

We have updated  the text in Results section, line 117-118, page 4.

-Please define the BRAF therapy received by the patients. Please discuss if a concentration dependence was examined and the impact of higher or lower BRAF therapies.

Mel KD, Mel JL and Mel JR patients received vemurafenib treatment. Patients achieved partial response as best response to therapy (Lai et al. 2012). MM90906, MM90911 and MM040111 patients did not received BRAF inhibitor therapy. We now included the response to vemurafenib treatment data in Table 1.

- Please provide further discussion/explanation on what it biologically means that only vimentin expression was increased in all cell lines tested.

Importantly, all investigated vemurafenib pre-treatment cell line lost their E-cadherin expression. In many cases one hallmark of EMT is the loss of E-cadherin expression (Pearlman et al. 2017). It was already described that oncogenic B-RAFV600E signaling could induce E-cadherin repression and also enhanced melanoma cell invasion (Boyd et al. 2013). Of note, it has also been described that BRAF inhibitor resistant melanoma cells also undergo EMT (Paulitschke et al. 2015; Wang et al. 2015). However Cordaro et al also found that dabrafenib resistant cells increase their motility, despite that EMT markers expression remains unvaried (E-cadherin, Snail) or even decreased (Twist) which suggest a “distinct active EMT-like” process adopted by melanoma cells under drug exposure (Cordaro et al. 2017). Our results further support this finding, since posttreatment cell lines showed considerable variability in expression changes of EMT markers. However further studies are needed to properly evaluate these findings. We have also further implemented Discussion section (line 182-184, page 9).

References:

Boyd, S. C., B. Mijatov, G. M. Pupo, S. L. Tran, K. Gowrishankar, H. M. Shaw, C. R. Goding, R. A. Scolyer, G. J. Mann, R. F. Kefford, H. Rizos, and T. M. Becker. 2013. 'Oncogenic B-RAF(V600E) signaling induces the T-Box3 transcriptional repressor to repress E-cadherin and enhance melanoma cell invasion', J Invest Dermatol, 133: 1269-77.

Cordaro, F. G., A. L. De Presbiteris, R. Camerlingo, N. Mozzillo, G. Pirozzi, E. Cavalcanti, A. Manca, G. Palmieri, A. Cossu, G. Ciliberto, P. A. Ascierto, S. Travali, E. J. Patriarca, and E. Caputo. 2017. 'Phenotype characterization of human melanoma cells resistant to dabrafenib', Oncol Rep, 38: 2741-51.

Lai, F., C. C. Jiang, M. L. Farrelly, X. D. Zhang, and P. Hersey. 2012. 'Evidence for upregulation of Bim and the splicing factor SRp55 in melanoma cells from patients treated with selective BRAF inhibitors', Melanoma Res, 22: 244-51.

Paulitschke, V., W. Berger, P. Paulitschke, E. Hofstatter, B. Knapp, R. Dingelmaier-Hovorka, D. Fodinger, W. Jager, T. Szekeres, A. Meshcheryakova, A. Bileck, C. Pirker, H. Pehamberger, C. Gerner, and R. Kunstfeld. 2015. 'Vemurafenib resistance signature by proteome analysis offers new strategies and rational therapeutic concepts', Mol Cancer Ther, 14: 757-68.

Pearlman, R. L., M. K. Montes de Oca, H. C. Pal, and F. Afaq. 2017. 'Potential therapeutic targets of epithelial-mesenchymal transition in melanoma', Cancer Lett, 391: 125-40.

Wang, J., S. K. Huang, D. M. Marzese, S. C. Hsu, N. P. Kawas, K. K. Chong, G. V. Long, A. M. Menzies, R. A. Scolyer, S. Izraely, O. Sagi-Assif, I. P. Witz, and D. S. B. Hoon. 2015. 'Epigenetic changes of EGFR have an important role in BRAF inhibitor-resistant cutaneous melanomas', J Invest Dermatol, 135: 532-41.

Wong, Deborah J. L., Lidia Robert, Mohammad S. Atefi, Amanda Lassen, Geetha Avarappatt, Michael Cerniglia, Earl Avramis, Jennifer Tsoi, David Foulad, Thomas G. Graeber, Begonya Comin-Anduix, Ahmed Samatar, Roger S. Lo, and Antoni Ribas. 2014. 'Antitumor activity of the ERK inhibitor SCH772984 [corrected] against BRAF mutant, NRAS mutant and wild-type melanoma', Molecular cancer, 13: 194-94.

Round 2

Reviewer 1 Report

This reviewer thanks the Authors for considering comments the improvement of data presentation.